# DINOReg: Strong Point Cloud Registration with Vision Foundation Model

## Abstract

Point cloud registration is a fundamental task in 3D computer vision. Most existing methods rely solely on geometric information for feature extraction and matching. Recently, several studies have incorporated color information from RGB-D data into feature extraction. Although these methods achieve remarkable improvements, they have not fully exploited the abundant texture and semantic information in images, and the feature fusion is performed in an image-lossy manner, which limit their performance. In this paper, we propose DINOReg, a registration network that sufficiently utilizes both visual and geometric information to solve the point cloud registration problem. Inspired by advances in vision foundation models, we employ DINOv2 to extract informative visual features from images, and fuse visual and geometric features at the patch level. This design effectively combines the rich texture and global semantic information extracted by DINOv2 with the detailed geometric structure information captured by the geometric backbone. Additionally, a mixed positional embedding is proposed to encode positional information from both image space and point cloud space, which enhances the model's ability to perceive spatial relationships between patches. Extensive experiments on the RGBD-3DMatch and RGBD-3DLoMatch datasets demonstrate that our method achieves significant improvements over state-of-the-art geometry-only and multi-modal registration methods, with a 14.2% increase in patch inlier ratio and a 15.7% increase in registration recall. The code is publicly available at [Anonymous] (provided in the supplementary material).

## 1 Introduction

Point cloud registration is an important task in 3D vision and serves as the foundation for applications such as 3D reconstruction and pose estimation. Given two point clouds, the goal of point cloud registration is to estimate a transformation that aligns their overlapping regions.

Feature-based methods are widely used for point cloud registration. They extract point-wise local features using descriptors, match them to establish correspondences, and estimate the transformation with algorithms such as SVD or RANSAC. Feature descriptors are crucial in this pipeline. Traditional methods rely on hand-crafted descriptors (Rusu et al., 2009; Salti et al., 2014), which calculate the statistical features of local geometric structure. With advances in deep learning, learning-based descriptors (Qi et al., 2017; Deng et al., 2018; Choy et al., 2019; Xu et al., 2021) have been proposed. These methods use MLPs or 3D convolutions to extract local feature representations, giving them strong ability to capture geometric information.

Recent research on feature-based registration has increasingly focused on transformer-based networks. Transformer (Vaswani et al., 2017) has strong ability to capture global context and long-range dependencies, offering a significant advantage over learning-based local feature descriptors. Predator (Huang et al., 2021) uses cross-attention for inter-frame information aggregation, enabling overlap prediction and reliable keypoint detection. CoFiNet (Yu et al., 2021) integrates interlaced self- and cross-attention layers with a coarse-to-fine matching strategy to obtain high-quality correspondences. GeoTransformer (Qin et al., 2022) introduces geometric structure embeddings to encode relative positional information between patches, allowing self-attention to model spatial relationships between patches. These methods have achieved remarkable progress, however, they rely solely on geometric information. When the overlap between two point clouds is extremely low or

Figure 1: Our method takes image-point cloud data pairs as input, extracting powerful multi-modal features for matching and accurate point cloud registration.

the geometric structures are ambiguous, such methods exhibit significant performance degradation, limiting their applicability.

To address this challenge, several studies have explored the use of visual information. These methods typically enhance local point cloud features with color data obtained from RGB-D inputs, resulting in more distinctive point-wise representations (Wang et al., 2022; Yuan et al., 2023; Yu et al., 2024b). ColorPCR (Mu et al., 2024) has achieved state-of-the-art performance in colored point cloud registration by incorporating point-wise color values into 3D convolutional layers. However, all existing methods make limited use of image information. Since images and point clouds have significantly different data distributions, simply colorizing points (Mu et al., 2024) only utilizes a fraction of the pixels and discards the dense texture and semantic information contained in images. Furthermore, using 3D convolutional layers to process image features (Yuan et al., 2023; Mu et al., 2024) raises concerns about preserving the structural information of the images, which inevitably leads to information loss and hinders effective fusion between geometric and visual features.

With the development of large models, the concept of the vision foundation model has been proposed. These models are trained on large-scale image datasets collected from the Internet in a self-supervised manner (Radford et al., 2021; Caron et al., 2021), giving them a strong ability to extract general-purpose features from images. Owing to their capacity and generalization in capturing global context and semantic information, pretrained vision foundation models can serve as powerful backbones for downstream tasks such as segmentation (Li et al., 2023) and depth estimation (Yang et al., 2024). Motivated by these advances, vision foundation models hold strong potential to provide rich and informative visual features for multi-modal point cloud registration.

In this paper, we propose DINOReg, a strong point cloud registration network for image–point cloud data, as shown in Fig. 1. We leverage the powerful vision foundation model DINOv2 (Oquab et al., 2023) to extract visual features, effectively capturing the rich texture and semantic information present in image data. To integrate the visual and geometric features, we design a patch fusion strategy. Specifically, we extract patch level features from visual and geometric backbones respectively, and spatially align the visual and geometric patches through mapping determination. As a result, each patch is associated with high-level and informative multi-modal features, which are subsequently fused using feed-forward network and attention modules. This enables our method to fully exploit both the texture and semantic information from images and the local geometric structure information from point clouds, producing distinctive representations for each patch.

Observing the limitations of geometric structure embeddings in providing positional information to the self-attention mechanism, we propose a mixed positional embedding. By encoding both the mapped pixel positions and the relative geometric structures of each patch, our method can capture spatial relationships between patches in both 2D and 3D space. Injecting regularly distributed pixel positions into the attention score calculation in a more tight manner also alleviates the drawbacks of geometric structure embeddings, effectively enhancing the model's ability to accurately capture spatial relationships and leading to improved matching and registration performance.

Since there are no suitable benchmarks for comprehensively evaluating registration methods using image–point cloud data, we construct the RGBD–3DMatch & RGBD-3DLoMatch datasets. Following the processing strategy of the 3DMatch & 3DLoMatch datasets (Huang et al., 2021), which are commonly used in previous geometry-only studies, we re-sample the original 3DMatch scanning data to generate image–point cloud data pairs with varying overlaps. Experimental results show that our method achieves state-of-the-art performance and surpasses previous approaches by a large margin. In conclusion, our main contributions are as follows:

- We explore the use of vision foundation model in point cloud registration and propose a novel network DINOReg, which fully exploits both image data and point cloud data.

- We design a spatial mapping and window aggregation strategy to effectively fuse multi-modal features at the patch level, producing highly distinctive representations.

- We propose a mixed positional embedding to enhance the model's ability to perceive spatial relationships between patches, which effectively improves the performance of feature matching and registration.

- We construct RGBD-3DMatch & RGBD-3DLoMatch datasets for evaluation using image-point cloud data. Comprehensive experiments demonstrate the superiority of our method.

## 2 RELATED WORK

**Feature-based point cloud registration**. Feature-based methods are widely used for solving registration tasks. They use hand-crafted descriptors (Rusu et al., 2009; Salti et al., 2014) or learning-based descriptors (Deng et al., 2018; Choy et al., 2019; Ao et al., 2021) to extract local feature representations for each point, then match them based on feature similarity. Due to the presence of outliers, a robust estimator is typically used for transformation estimation. RANSAC (Fischler & Bolles, 1981) is a commonly adopted estimator, and methods focus on outlier rejection (Choy et al., 2020; Bai et al., 2021; Jiang et al., 2023) and geometric consistency (Chen et al., 2022; Zhao et al., 2025; Zhang et al., 2025) also demonstrate effectiveness. We follow the pipeline of feature-based method, providing powerful multi-modal feature representations for matching and transformation estimation.

**Transformer-based registration network**. Transformer-based methods leverage the global context aggregation capability of attention modules to learn more distinctive representations. These methods typically first extract local geometric features using 3D convolutional networks (Wang et al., 2019; Yang et al., 2018; Choy et al., 2019; Thomas et al., 2019), then use self-attention and cross-attention mechanisms to aggregate both intra-frame and inter-frame information (Huang et al., 2021; Yu et al., 2021; 2023; 2024c;a; Zeng et al., 2025). In transformer-based registration networks, positional embedding has proven to be crucial, which enables the network to model spatial relationships between points (Qin et al., 2022; Yang et al., 2022). In our work, we adopt attention mechanisms for global context aggregation and feature fusion, and propose a mixed positional embedding for the network to more effectively learn spatial relationships.

**Multi-modal point cloud registration**. Multi-modal registration methods integrate visual information into feature extraction, which addresses the limitations of geometric features and produces more informative representations. These methods use visual information to enhance feature learning (El Banani et al., 2021; El Banani & Johnson, 2021), or separately extract visual and geometric local features and fuse them to obtain multi-modal representations for each point (Wang et al., 2022; Zhang et al., 2022; Yuan et al., 2023; Jiang et al., 2025). PointMBF (Yuan et al., 2023) uses ResNet He et al. (2016) and KPConv Thomas et al. (2019) to extract features from images and point clouds respectively, and exchanges the features at each layer in a bidirectional manner for multi-scale fusion. Our main competitor, ColorPCR (Mu et al., 2024), enhances 3D convolutional features with additional point-wise color information, resulting in more distinctive local features. However, these methods make limited use of images, and point-wise fusion operations based on 3D convolutions tend to cause the loss of structural and semantic information. In contrast, our method fully exploits the texture and semantic information from images with vision foundation model, and effectively fuses visual and geometric information at the patch level, leading to remarkable performance.

## 3 METHOD

The pipeline of our DINOReg is illustrated in Fig. 2. Given a pair of point clouds $\mathbf{P}, \mathbf{Q} \in \mathbb{R}^{N \times 3}$ and their corresponding images $\mathbf{I}^P, \mathbf{I}^Q \in \mathbb{R}^{H \times W \times 3}$, the model first extracts multi-modal patch features via visual and geometric backbones. Then, it determines spatial mappings between visual and geometric patches and fuses the corresponding multi-modal features. Attention modules are employed for global context aggregation and further fusion, producing distinctive fused features for feature matching. The transformation between the two point clouds is then estimated with the extracted matches. In the following sections, we will introduce the proposed modules in detail.

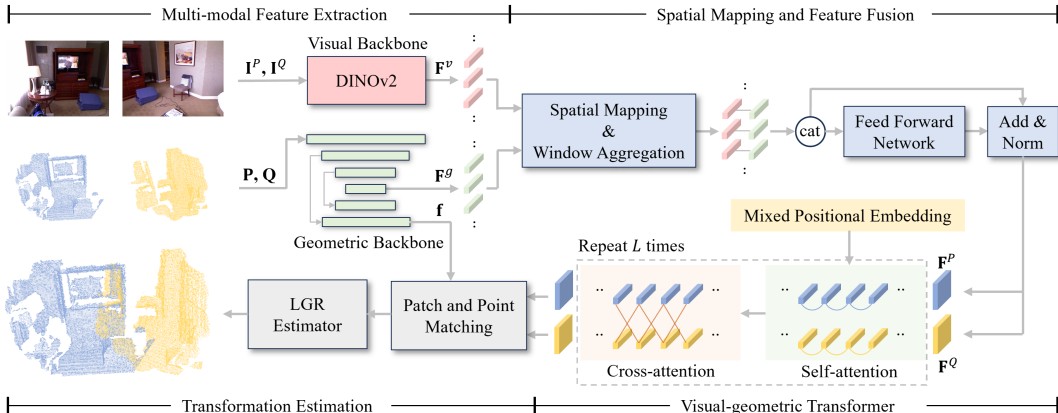

Figure 2: Pipeline of our method. Our method extracts multi-modal features from image and point cloud data, and fuses them at the patch level to obtain distinctive representations for matching. The transformation between two point clouds is then estimated using the matches.

## 3.1 Multi-modal Feature Extraction

We adopt both a visual backbone and a geometric backbone to extract features from image and point cloud data, respectively. The extracted multi-modal features are then fed into subsequent modules.

**Visual backbone.** Pretrained vision foundation models have proven effective as backbones for downstream tasks such as depth estimation (Yang et al., 2024) and feature matching (Jiang et al., 2024). Therefore, we adopt DINOv2 (Oquab et al., 2023) as the visual backbone to extract patch features from each input image. We use the patch features from the last layer of the ViT module as the output $\mathbf{F}^v$. Each patch feature represents a $14 \times 14$ pixels region of the image and contains abundant texture and semantic information.

**Geometric backbone.** To obtain geometric features with a distribution similar to the visual patch features, we adopt KPConv-FPN (Thomas et al., 2019) structure as the geometric backbone to extract multilevel local geometric features from the point clouds. The output features from the last downsampling level are regarded as geometric patch features $\mathbf{F}^g$, which represent local regions of the point cloud and are used for feature fusion and patch matching. Features from the first downsampling level are treated as point features $\mathbf{f}$ and are used for point matching.

## 3.2 Spatial Mapping and Feature Fusion

After multi-modal feature extraction, we obtain visual and geometric patch features from the image and point cloud data. The geometric features contain structural information about local point cloud regions, while the visual features contain both the texture of local image regions and global semantic context. Our goal is to effectively fuse visual and geometric features to extract a distinctive and informative representation for each patch. To this end, we first need to determine the spatial mappings between the visual and geometric patches to align their spatial positions, as shown in Fig. 3.

**Spatial mapping determination.** Since each visual patch corresponds to a local region in the original image, we remove the [CLS] token and reshape the visual patch features into a $H' \times W'$ feature map $\mathbf{F}^{v\_map}$. This grid map indicates the position of each visual patch on the image plane. We then leverage the calibration parameters to project the 3D coordinates of geometric patches $\mathbf{X}^{pc}$ onto image plane, resulting in a set of 2D pixel positions $\mathbf{X}^{img}$ corresponding to each geometric patch. Using the scaled pixel positions $\hat{\mathbf{X}}^{img}$, we index the visual feature map to retrieve the corresponding visual patch feature for each geometric patch. This establishes the spatial mappings between the patches of two modalities. Further details can be found in the Appendix A.1.

With the spatial mappings established, each geometric patch is associated with both its own geometric feature and the feature from its corresponding visual patch. However, due to different data resolutions and inherent mapping inaccuracies, a geometric patch might be incorrectly mapped to a

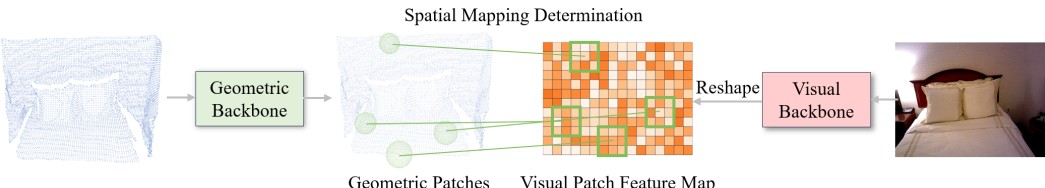

Figure 3: Illustration of spatial mapping and window aggregation. Geometric patches (we choose four as examples) are mapped to pixel positions, and visual patch features around the mapped positions are aggregated to obtain the associated visual features of each geometric patch.

patch adjacent to its true corresponding visual patch. Furthermore, because of the different distribution properties of 2D and 3D space, two patches that are neighbors in 3D space may be mapped to two patches that are not adjacent in the 2D space. This results in many visual patches having no valid mappings, particularly when geometric patches are quite sparse. Consequently, these unmapped visual patches are discarded, resulting in an inadequate utilization of visual information. To address these issues, we leverage a window aggregation strategy.

**Window aggregation**. For each geometric patch, we extract a $K \times K$ window of visual patch features centered at its mapped location in the visual feature map. These features are then aggregated into the final corresponding visual feature through a convolution layer with a kernel size of K:

$$\mathbf{F}_i^{v\_win} = \sum_{p=1}^{K} \sum_{q=1}^{K} \mathbf{W}_{p,q} \mathbf{F}_{u_i+p-r,v_i+q-r}^{v\_map} + \mathbf{b}_{p,q}, \tag{1}$$

where $r = \lfloor K/2 \rfloor + 1$, $u_i$ and $v_i$ are the scaled pixel positions of geometric patches, $\mathbf{W}_{p,q}$ and $\mathbf{b}_{p,q}$ are the weight and bias of convolution layer, respectively. Using window aggregation, each patch is associated with a larger region of visual features, mitigating the negative effects of both mapping inaccuracies and visual information loss. We then concatenate the geometric feature $\mathbf{F}_i^g$ and visual feature $\mathbf{F}_i^{v\_win}$ for each patch and use a feed-forward network to fuse the multi-modal features in latent space. As a result, each patch is assigned with a fused feature $\mathbf{F}_i$, which provides an informative geometric and visual representation.

By performing feature fusion at patch level, our method effectively leverages the powerful semantic and global context aware capabilities of DINOv2. Concurrently, the local geometric features extracted via 3D convolution network complement the DINO features by providing detailed spatial structural information, which is challenging to capture accurately through image. This enables the fused features to have both discriminative global semantics and distinctive local structural details.

### 3.3 VISUAL-GEOMETRIC TRANSFORMER

After obtaining the fused features for each patch, we employ attention modules to further aggregate global context. Following previous studies (Yu et al., 2021; Qin et al., 2022), we perform self-attention and cross-attention in an interlaced manner. This structure enables interaction between intra-frame and inter-frame features, facilitating a deeper fusion of visual and geometric information.

In self-attention mechanism, positional embedding is crucial for registration tasks. Since patches have inherent relative spatial relationships, positional embedding allows the model to recognize these relationships, thereby facilitating the modeling of intra-frame structural information and inter-frame geometric consistency. Previous study proposed a geometric structure embedding (Qin et al., 2022), which encodes the relative distances and relative angles between patches and incorporates them into the attention score calculation:

$$e_{ij} = \frac{(\mathbf{W}^Q \mathbf{F}_i)^T \mathbf{W}^K \mathbf{F}_j + (\mathbf{W}^Q \mathbf{F}_i)^T \mathbf{W}^R \mathbf{r}_{ij}}{\sqrt{d}}, \tag{2}$$

where $d$ is the feature dimension, $\mathbf{W}^Q, \mathbf{W}^K, \mathbf{W}^R$ are the projection matrices for queries, keys and geometric embeddings, $\mathbf{r}_{ij}$ is the embedding calculated by the relative distance and angles between patches $i, j$. This approach enables the model to perceive spatial structures among patches. However, it operates by additively modifying the attention scores rather than directly participating in the

computation of the dot product between queries and keys. When the dot product between queries and keys is large, the influence of this additive term on the attention scores becomes weak. Moreover, due to the irregularity of point cloud data, the distribution of relative distances and angles varies across different samples. Consequently, learned embeddings that work well for small point clouds may not be suitable for larger ones, hindering the model's ability to learn spatial relationships accurately. To address these limitations and improve the model's capacity to capture spatial structures between patches, we propose a mixed positional embedding that enables the model to simultaneously perceive positional information from both image space and point cloud space.

**Mixed positional embedding**. Due to the perspective effect, the relative positions of patches in 2D image space differ from in 3D space. Incorporating 2D pixel positions allows the model to recognize this property, leading to better structural awareness. Moreover, unlike point clouds, images have a regular structure, and the normalized positions have a consistent distribution across different samples. This helps the model learn spatial relationships more effectively. We first normalize the 2D pixel positions of each patch and use MLP to project them into high-dimensional embeddings $\mathbf{p}$. Following rotary positional embedding approach (Su et al., 2024), we then calculate rotary matrices using $\mathbf{p}$:

$$\mathbf{R}(\mathbf{p}) = \begin{pmatrix} \mathbf{R}_\theta(p_0) & & 0 \\ & \ddots & \\ 0 & & \mathbf{R}_\theta(p_{d/2}) \end{pmatrix}, \quad \mathbf{R}_\theta(p) = \begin{pmatrix} \cos p & -\sin p \\ \sin p & \cos p \end{pmatrix}. \tag{3}$$

By applying rotary matrices to the queries, keys, and geometric embeddings, relative positional information can be incorporated. Since the calculation of geometric embedding has a $O(n^2 d^2)$ complexity with patch number $n$ and dimension $d$, we adopt a shared formulation (Qin et al., 2023) instead of computing it in each self-attention layer with independent projection matrices:

$$\hat{\mathbf{r}}_{ij} = \mathbf{W}^R \phi(\mathbf{r}_{ij}), \tag{4}$$

where $\phi$ is the LeakyReLU function, and $\mathbf{W}^R$ is shared across all layers. This formulation allows mixed embeddings to be pre-computed and cached, as they are identical for all layers. Since image relative positional information plays distinct roles in the interactions between queries and keys, and between queries and geometric embeddings, we assign separate learnable $\mathbf{p}$ to each of them:

$$e_{ij} = \frac{[\mathbf{R}(\mathbf{p}_i)\mathbf{W}^Q\mathbf{F}_i]^T[\mathbf{R}(\mathbf{p}_j)\mathbf{W}^K\mathbf{F}_j] + [\mathbf{R}(\mathbf{p}'_i)\mathbf{W}^Q\mathbf{F}_i]^T[\mathbf{R}(\mathbf{p}'_j)\hat{\mathbf{r}}_{ij}]}{\sqrt{d}}. \tag{5}$$

As a result, both 2D and 3D positional information are injected into self-attention modules, enabling the model to learn comprehensive relative spatial relationships.

## 3.4 Training and Transformation Estimation

We adopt the coarse-to-fine matching strategy (Yu et al., 2021) for feature matching. We first match the patches between the source and target point clouds using patch features refined by attention modules. Subsequently, we perform local point-wise matching within each patch pair using the dense features extracted by geometric backbone. Following previous studies (Qin et al., 2022; Mu et al., 2024), we use the overlap-aware circle loss (Qin et al., 2022) for supervising patch matching, and a negative log-likelihood loss for supervising point matching. During training, the parameters of DINOv2 are frozen.

Our model produces high-quality point matches that can be used for transformation estimation by estimators such as RANSAC. We follow the previous studies (Qin et al., 2022; Yu et al., 2023; Mu et al., 2024) to employ the local-to-global registration approach (Qin et al., 2022), a fast and robust estimator that is commonly used in coarse-to-fine matching methods.

## 4 Experiment

In this section, we evaluate our method on both indoor and outdoor datasets, comparing it with state-of-the-art registration methods. We also provide comprehensive ablation studies to validate the effectiveness of our designs.

Table 1: Evaluation results on RGBD-3DMatch & RGBD-3DLoMatch datasets.

| | RGBD-3DMatch | | | | RGBD-3DLoMatch | | | | Time(s) |
|---|---|---|---|---|---|---|---|---|---|
| | PIR(%) | IR(%) | FMR(%) | RR(%) | PIR(%) | IR(%) | FMR(%) | RR(%) | |
| Predator (Huang et al., 2021) | - | 26.6 | 85.2 | 78.8 | - | 7.7 | 45.8 | 32.9 | 0.879 |
| CoFiNet (Yu et al., 2021) | 52.1 | 40.6 | 94.2 | 84.3 | 20.2 | 15.9 | 65.5 | 45.8 | 0.128 |
| GeoTransformer (Qin et al., 2022) | 64.0 | 48.4 | 95.7 | 87.5 | 27.5 | 20.8 | 69.7 | 51.0 | 0.172 |
| PEAL-3D (Yu et al., 2023) | 72.6 | 53.8 | 93.8 | 88.6 | 37.0 | 25.9 | 64.5 | 53.8 | 1.592 |
| ColorPCR (Mu et al., 2024) | 64.6 | 48.0 | 96.9 | 89.6 | 29.6 | 21.5 | 75.2 | 57.2 | 0.213 |
| DINOReg (*ours*) | **74.6** | 54.9 | 99.6 | **96.2** | 43.8 | 30.6 | 90.4 | 72.9 | 0.314 |
| DINOReg-Super (*ours*) | **74.6** | **55.2** | **99.9** | 96.0 | **44.7** | **31.4** | **90.7** | **74.9** | 0.407 |

## 4.1 RESULTS ON RGBD-3DMATCH & RGBD-3DLOMATCH DATASETS

**Dataset**. 3DMatch & 3DLoMatch indoor datasets (Huang et al., 2021) are widely used for registration evaluation, which are sampled from original 3DMatch scanning data (Zeng et al., 2017). Based on them, Color3DMatch dataset (Mu et al., 2024) colorizes point clouds by performing scene reconstruction. However, both 3DMatch and Color3DMatch are unsuitable for evaluating methods using image data, as they merge 50-frames point clouds to generate samples, making it difficult to determine the corresponding images for each sample. To this end, we re-sample the 3DMatch scanning data and create the RGBD-3DMatch and RGBD-3DLoMatch datasets for evaluation. We follow the processing strategy used in 3DMatch & 3DLoMatch but remove the merging operation. This produces data pairs that include both point clouds and their corresponding images. Moreover, removing the merging operation results in substantially smaller overlap and introduces more realistic data noise, making the benchmarks more challenging. Details can be found in Appendix A.2.

**Baselines**. We select keypoint matching method Predator (Huang et al., 2021), and coarse-to-fine matching methods CoFiNet (Yu et al., 2021), GeoTransformer (Qin et al., 2022), PEAL-3D (Yu et al., 2023), and ColorPCR (Mu et al., 2024). We evaluate two versions of our method. The standard DINOReg uses DINOv2-small as the visual backbone, while keeping the channel and depth configurations of the geometric backbone and attention modules consistent with baselines (Qin et al., 2022; Mu et al., 2024). DINOReg-Super uses DINOv2-base as the visual backbone and adopts larger channel sizes for the fusion layers and attention modules. For all coarse-to-fine methods, the LGR estimator is used for transformation estimation. For the keypoint matching method Predator, we follow its original pipeline for transformation estimation.

**Metrics**. Following previous studies (Qin et al., 2022), we use patch inlier ratio (PIR), inlier ratio (IR), feature matching recall (FMR) and registration recall (RR) to evaluate the performance. PIR is the fraction of patch matches that have actual overlap. IR is the fraction of point matches whose residuals are below a threshold (i.e. 0.1 m). FMR is the fraction of data pairs whose IR is above a certain threshold (i.e. 5%). RR is the fraction of data pairs whose RMSE is smaller than a threshold (i.e. 0.2 m). We also evaluate the average total time cost on RGBD-3DMatch dataset.

**Training details**. We train our model for 20 epochs on RGBD-3DMatch dataset with a batch size of 1 and a weight decay of $10^{-6}$. We use a single NVIDIA 3090 for implementation. The learning rate is initialized at $10^{-4}$ and decays exponentially by 0.05 per epoch. We also retrain all baseline methods on RGBD-3DMatch dataset for a fair comparison.

**Results**. The results are shown in Table 1. Our method achieves state-of-the-art performance on both datasets, with significant improvements over previous methods. Compared to geometric-only method GeoTransformer, we improve PIR by 16.3% and RR by 21.9% on RGBD-3DLoMatch dataset. And compared to the state-of-the-art colored point cloud registration method ColorPCR, we improve PIR by 14.2% and RR by 15.7%. These results demonstrate the strong performance of our method. PEAL-3D uses transformation prior estimated by GeoTransformer to identify overlapping points and iteratively refine the features. This effectively improves PIR, but dependence on the accuracy of transformation prior makes it challenging to achieve higher RR. ColorPCR injects point-wise color information into KPConv-FPN to extract enhanced local features. Although effective, it cannot fully exploit the dense texture and semantic information in image data, and is sensitive to the mapping accuracy between pixels and points. By leveraging the powerful features extracted by vision foundation model, and fusing multi-modal features at patch level, our method sufficiently utilizes image information and maintains strong robustness against inaccurate mappings, resulting in a significant performance advantage. Notably, when using DINOv2-base as the backbone, our method

Table 2: Evaluation results on KITTI dataset.

|  | PIR(%) | IR(%) | RRE(°) | RTE(cm) | RR(%) |
|---|---|---|---|---|---|
| Predator | - | 25.9 | 0.38 | 16.1 | 96.6 |
| CoFiNet | 73.5 | 55.5 | 0.40 | 11.7 | 98.7 |
| GeoTransformer | 75.3 | 57.4 | 0.41 | 11.1 | 98.9 |
| ColorPCR | 73.2 | 58.2 | 0.42 | 12.2 | 98.6 |
| DINOReg (*ours*) | **78.4** | **60.2** | **0.37** | **9.8** | **99.3** |

further improves PIR by 0.9% and RR by 2.0% on RGBD-3DLoMatch dataset. This indicates that our method can benefit from more powerful vision foundation models.

## 4.2 RESULTS ON KITTI DATASET

**Dataset**. KITTI is an outdoor dataset (Geiger et al., 2012), containing images captured by camera and point clouds collected by LiDAR. We following previous studies to use the sequences 0-5 for training, 6-7 for validation and 8-10 for testing. Since the LiDAR's field of view is much larger than that of the camera, we crop the point clouds to retain only the points within camera's view, producing image-point cloud pairs for evaluation. As the cropped point clouds have a much smaller overlap, we remove data pairs with an overlap ratio lower than 5%.

**Metrics**. Following previous studies (Qin et al., 2022), we report relative rotation error (RRE), relative translation error (RTE) and registration recall (RR). RR is the fraction of data pairs whose RRE and RTE are both below the thresholds (i.e. 5° and 2 m). We report the average RRE and RTE for successfully registered pairs. We also report PIR and IR for evaluating matching performance.

**Training details**. We train our model for 60 epochs with a batch size of 1 and a weight decay of $10^{-6}$. The learning rate is initialized at $10^{-4}$ and decays exponentially by 0.05 every 4 epochs. We also retrain all baseline methods on the cropped KITTI dataset for a fair comparison.

**Results**. As shown in Table 2, our method achieves the best performance. KITTI is a less challenging dataset where previous methods have already achieved near-saturated performance. By leveraging image information, our method improves the quality of patch matches and point matches, leading to lower RRE and RTE. Notably, ColorPCR performs worse than GeoTransformer. This is mainly because LiDAR point clouds have no accurate pixel-point mappings with images, and the mapping errors lead to wrong point-wise colorization. In contrast, our method is robust to mapping errors, thereby achieving improved performance.

## 4.3 ABLATION STUDIES

To demonstrate the effectiveness of our designs, we conduct ablation studies on the key components of our model including (a) feature fusion, (b) window aggregation, and (c) positional embedding. The results are shown in Table 3. In addition, we evaluate our method's performance under inaccurate mappings to further demonstrate its robustness.

**Feature fusion**. We evaluate performance under four settings. For geometric feature only setting, visual backbone and fusion layers are removed, which is equivalent to GeoTransformer structure combined with our proposed mixed positional embedding. In the visual feature only setting, we use only visual patch features $\mathbf{F}^{v\text{-}win}$ for attention aggregation and patch mapping. Additionally, we test a variation that the feed-forward network is removed, and the concatenated features are directly fed into the attention modules. As shown in (a.1)-(a.4), using single-modality features leads to lower performance. Benefited from the powerful representation of DINO features, visual only setting outperforms the geometric only setting. However, by incorporating local geometric structure information through feature fusion, our method achieves an improvement of 8.2% in PIR and 5.5% in RR on RGBD-3DLoMatch. Furthermore, using a feed-forward network to initially fuse features before attention modules increases PIR by 3.1% and RR by 1.4%, demonstrating its importance.

**Window aggregation**. We evaluate the performance under different sizes of window. As shown in (b.1)-(b.3), aggregating neighboring visual patch features improves PIR by 1.5% and RR by 1.3% compared to one-to-one mapping. However, increasing the window size to $5 \times 5$ leads to a slight

Table 3: Ablation studies on RGBD-3DMatch & RGBD-3DLoMatch datasets.

|  | RGBD-3DMatch | | | | RGBD-3DLoMatch | | | |
| --- | --- | --- | --- | --- | --- | --- | --- | --- |
|  | PIR(%) | IR(%) | FMR(%) | RR(%) | PIR(%) | IR(%) | FMR(%) | RR(%) |
| (a.1) Geometric feature only | 65.5 | 49.5 | 97.1 | 91.4 | 32.1 | 23.4 | 77.3 | 60.7 |
| (a.2) Visual feature only | 65.0 | 50.0 | 98.9 | 93.3 | 35.6 | 26.4 | 86.9 | 67.4 |
| (a.3) Concatenation | 71.7 | 53.1 | 99.4 | 95.6 | 40.7 | 29.1 | 89.8 | 71.5 |
| (a.4) Feed-forward network* | **74.6** | **54.9** | **99.6** | **96.2** | **43.8** | **30.6** | **90.4** | **72.9** |
| (b.1) $K = 1$ (one-to-one) | 73.6 | 54.2 | 99.0 | 95.3 | 42.3 | 29.6 | 88.3 | 71.6 |
| (b.2) $K = 3$* | **74.6** | **54.9** | **99.6** | **96.2** | **43.8** | **30.6** | **90.4** | **72.9** |
| (b.3) $K = 5$ | 74.0 | 54.6 | 99.2 | 95.8 | 43.0 | 30.5 | 89.7 | 72.3 |
| (c.1) w/o embedding | 68.4 | 50.3 | 99.2 | 95.1 | 39.7 | 27.9 | 88.6 | 69.7 |
| (c.2) Geometric embedding | 73.3 | 54.2 | 99.4 | 95.7 | 42.0 | 29.7 | 88.9 | 70.6 |
| (c.3) Mixed embedding* | **74.6** | **54.9** | **99.6** | **96.2** | **43.8** | **30.6** | **90.4** | **72.9** |

Table 4: Evaluation results under different levels of mapping noise on RGBD-3DLoMatch dataset.

|  | noise with $\sigma = 5$ | | | | noise with $\sigma = 10$ | | | |
| --- | --- | --- | --- | --- | --- | --- | --- | --- |
|  | PIR(%) | IR(%) | FMR(%) | RR(%) | PIR(%) | IR(%) | FMR(%) | RR(%) |
| ColorPCR | 28.4 | 20.6 | 72.6 | 53.7 | 26.8 | 19.3 | 69.2 | 48.7 |
| DINOReg (*ours*) | 43.4 | 30.3 | 90.1 | 72.6 | 42.2 | 29.6 | 89.0 | 71.6 |

performance drop. This is because on RGBD-3DMatch dataset, $3 \times 3$ window is sufficient to cover unmapped visual patches and relieve mapping inaccuracies. Further enlarging the window size does not incorporate additional visual patches, but instead tends to blur the assigned visual features of neighboring geometric patches, as a larger window increases the overlap of visual features between two neighbors. For applications, the window size should be chosen based on the distribution differences between visual and geometric patches.

**Positional embedding**. Positional embedding is crucial for enabling the model to perceive spatial relationships between patches. Although DINOv2 incorporates positional embedding in the ViT modules, the positional information is utilized by the attention modules indirectly, which is insufficient. As shown in (c.1)-(c.3), adding geometric structure embedding improves PIR by 2.3% and RR by 0.9%. Furthermore, our mixed positional achieves an additional improvement of 1.8% in PIR and 2.3% in RR, demonstrating its effectiveness. Notably, comparing (a.1) with the results of GeoTransformer reported in Table 1, our mixed positional embedding significantly boosts the performance of GeoTransformer, improving PIR by 4.6% and RR by 9.7% on RGBD-3DLoMatch dataset. This clearly highlights the significant advantages of the proposed mixed positional embedding.

**Mapping inaccuracies**. Our method performs feature fusion at the patch level and uses window aggregation to assign visual features to geometric patches. These bring our method strong robustness against mapping inaccuracies, which is crucial for practical applications. To make further demonstration, we evaluate the performance of our method and ColorPCR under inaccurate mappings by adding Gaussian noise to the mapped pixel positions. We set two standard variances $\sigma = 5$ and $\sigma = 10$ to represent low and high noise levels. As shown in Table 4, our method exhibits negligible degradation under low noise level and only slight degradation under severe noise. In contrast, ColorPCR suffers a significant performance drop due to incorrect point-wise colorization. In real-world scenarios, inaccurate mappings often occur due to rough calibration or differences in data distribution. Therefore, our method has strong robustness and reliability in handling practical tasks.

## 5 CONCLUSION

In this paper, we propose a strong point cloud registration network DINOReg. By effectively leveraging the abundant texture and semantic information extracted from vision foundation model and fusing it with local geometric information, our method significantly improves registration performance. Fusing multi-modal features at the patch level also provides strong robustness against mapping inaccuracies, enhancing the applicability of our method to practical tasks. Moreover, the proposed mixed positional embedding injects comprehensive relative positional information into the network, demonstrating remarkable performance improvement. In future work, we would like to explore the application of our method to a wider range of multi-modal tasks.

## REPRODUCIBILITY STATEMENT

We propose a novel point cloud registration network for image-point cloud data. For reproducibility, we provide details about dataset construction and model architecture in Appendix. Besides, we also provide the complete code of our method and data construction process in the supplemental material. We will also publicly release the code, datasets and pre-trained models to facilitate reproducibility.

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

# A   APPENDIX

## A.1   DETAILS ABOUT SPATIAL MAPPING

Given the 3D coordinates of geometric patches $\mathbf{X}^{pc}$, we first projected them onto image plane to obtain their 2D pixel positions $\mathbf{X}^{img}$. For RGBD-3DMatch and RGBD-3DLoMatch datasets, since the point clouds are generated from the depth maps, we can directly use the intrinsic matrix $\mathbf{K}$ for projection:

$$\begin{pmatrix} u_i \times s \\ v_i \times s \\ s \end{pmatrix} = \mathbf{K}\mathbf{X}_i^{pc}. \tag{6}$$

After remove the scale coefficient $s$ by normalization, 2D pixel positions $\mathbf{X}^{img}$ of each patch are determined. For KITTI dataset, since it uses LiDAR to collect point cloud data, we need to first transform $\mathbf{X}^{pc}$ to the camera coordinate system:

$$\mathbf{X}_i^{pc'} = \mathbf{R}\mathbf{X}_i^{pc} + \mathbf{t}, \tag{7}$$

where $\mathbf{R}$ and $\mathbf{t}$ are the calibrated rotation matrix and translation vector between camera and LiDAR coordinate systems. Subsequently, $\mathbf{X}^{img}$ can be determined by $\mathbf{K}$ in the same way. In this step, the invalid patches that locate out of the images are removed.

Given the image input size $H \times W$ and the output size of the visual patch feature map from DINOv2 $H' \times W'$, we scale and round $\mathbf{X}^{img}$ to obtain the corresponding locations on the grid map $\mathbf{F}^{v\text{-}map}$:

$$\hat{X}_u^{img} = \lfloor \frac{W'}{W} X_u^{img} \rfloor, \ \ \hat{X}_v^{img} = \lfloor \frac{H'}{H} X_v^{img} \rfloor. \tag{8}$$

Using $\hat{\mathbf{X}}^{img}$, we can index into $\mathbf{F}^{v\text{-}map}$ to retrieve corresponding visual patch features for each geometric patch.

Notably, we also apply the spatial mapping method to colorize the point clouds for ColorPCR. In the ColorPCR paper (Mu et al., 2024), the authors first reconstruct each scene using all samples from the 3DMatch scanning data. Then, each point cloud sample is aligned to the reconstructed scene using the pose annotations, and its points are colorized based on the points in the reconstructed scene. This process effectively reduces potential incorrect colorization by scene reconstruction. However, it requires complete scene samples and pose annotations, making it inapplicable for test-time and practical RGB-D registration tasks, where future frames and ground-truth data are unavailable. Therefore, we use the spatial mapping approach to colorize points for each individual sample, indexing color values from images via the mapped 2D pixel positions of dense points. This ensures a realistic and referable evaluation.

## A.2   DETAILS ABOUT CONSTRUCTION OF RGBD-3DMATCH & RGBD-3DLOMATCH

We note that there is currently a lack of suitable image–point cloud pair registration benchmarks. Existing registration datasets such as 3DMatch and Color3DMatch cannot be directly applied to image–point cloud registration tasks, as they merge point cloud data from 50 consecutive frames, making it difficult to determine the corresponding image data.

To address this issue, we re-sample the original 3DMatch scanning data to construct new RGBD-3DMatch & RGBD-3DLoMatch datasets for evluation. We removed scenes in 3DMatch that do not contain RGB images, resulting in 39 / 7 / 8 scenes for training / validation / testing, respectively. Following the sampling strategy of the 3DMatch & 3DLoMatch datasets, we sample the data every 50 frames or 100 frames for different scenes (same to 3DMatch & 3DLoMatch). Unlike the previous strategy, we no longer merge the 50 consecutive frames. Instead, we directly use the RGB and depth images from a single frame to construct the point cloud samples.

After obtaining the samples, we traverse possible data pairs and calculate the overlap ratio between point clouds, data pairs that have an overlap ratio $\geq 5\%$ are involved. During traversal, we also impose several restrictions.

**Group sampling**. For each scene, we group the samples into sets of 60, dividing the scene into multiple groups. And we perform pairwise traversal of the data pairs within each group. This

grouping strategy prevents scenes with a very large sample count $N$ from producing a factorial-scale amount of data pairs (on the order of $N!$), which could dominate the dataset distribution and cause the model to overfit to specific scenes.

**Validation set**. After removing the scenes without RGB images, we find that the validation set becomes too small. Therefore, we adjust the scene split between the training and validation sets to increase the number of data pairs in the validation set. Notably, the split of the test set remains unchanged.

**Test set**. For the test set, we follow the strategy of the 3DMatch & 3DLoMatch datasets, limiting the total sample counts of each scene. The number of samples per scene is limited to no more than 100, this further balances the amount of data pairs across scenes. Same to 3DMatch & 3DLoMatch datasets, we divide the data paris by overlap ratio, where RGBD-3DMatch contains pairs in the 30%–70% range, and RGBD-3DLoMatch contains pairs in the 10%–30% range.

In total, we generate 28.0k / 2.2k / 1.9k / 1.9k data pairs for the train, validation, RGBD-3DMatch, and RGBD-3DLoMatch sets, respectively. This construction method ensures that each point cloud sample has a corresponding single image. Besides, removing the merging operation introduces more realistic noise and results in smaller point cloud samples. Under the same overlap ratio, data pairs in our constructed datasets share fewer overlapping points than those in 3DMatch & 3DLoMatch, making the benchmarks considerably more challenging. Moreover, since the merging process is usually absent in practical applications, our datasets can better reflect the real situations.

## A.3 DETAILS OF MODEL ARCHITECTURE

Following previous studies (Qin et al., 2022), we adopt a 4-stage KPConv-FPN structure for the RGBD-3DMatch & RGBD-3DLoMatch datasets, and a 5-stage KPConv-FPN structure for the KITTI dataset. The output patch feature channels are 1,024 and 2,048, respectively. Before feature fusion, the channels of the geometric and visual patch features are both reduced to 256 for standard DINOReg and 512 for DINOReg-Super. The hidden size and output size of feed-forward network are 1,024 / 512 for standard DINOReg and 2,048 / 1,024 for DINOReg-Super.

For the standard DINOReg, we follow previous studies (Qin et al., 2022; Mu et al., 2024) and set $L = 3$ for the attention modules. Each attention layer has a hidden size of 256 with 4 attention heads. For DINOReg-Super, we increase the hidden size to 512 and the number of heads to 8 due to the enlarged feature channels of DINOv2-base.

## A.4 USAGE OF LARGE LANGUAGE MODELS

We use large language models only for polishing the writing and correcting grammatical errors, and no other usage is involved.

