# OpenReview forum: "DINOReg: Strong Point Cloud Registration with Vision Foundation Model"
_ICLR.cc/2026/Conference — ICLR 2026 Conference Withdrawn Submission_

### Official Review · Reviewer_cwDP · 2025-10-27

**Soundness:** 2
**Presentation:** 3
**Contribution:** 2
**Rating:** 4
**Confidence:** 4

**Summary:**

The paper introduces DINOReg, a novel point cloud registration method that integrates both visual and geometric information. While traditional methods primarily rely on geometric features, DINOReg leverages the DINOv2 vision foundation model to extract rich texture and semantic features from images, which are then fused with geometric features extracted from point clouds using the KPConv-FPN backbone. Key innovations include a spatial mapping strategy to align visual and geometric patches, followed by window aggregation to improve feature fusion and robustness against mapping inaccuracies. Additionally, a mixed positional embedding is introduced to capture spatial relationships between patches in both 2D and 3D spaces. The model employs a transformer-based attention mechanism, combining self-attention and cross-attention to aggregate global context and enhance multi-modal fusion. Evaluations on datasets such as RGBD-3DMatch and RGBD-3DLoMatch demonstrate significant improvements in patch matching, registration recall, and robustness to noise and mapping errors, showcasing the effectiveness of combining visual and geometric data for point cloud registration tasks.

**Strengths:**

1. It combines visual and geometric features, improving point cloud registration.
2. It is robust to mapping errors through window aggregation, handling noisy data well.
3. The mixed positional embedding enhances spatial awareness by capturing both 2D and 3D relationships.
Overall, it is an interesting attempt on cross modal task.

**Weaknesses:**

Here are a few potential weaknesses of DINOReg:
1. Dependency on Large Models: The method relies on the DINOv2 vision foundation model, which may require significant computational resources for training and inference, especially for larger datasets.
2. Mapping Inaccuracies: While the method is robust to some mapping errors, it may still struggle with extreme inaccuracies in the mapping between visual and geometric patches, affecting performance in challenging real-world scenarios.
3. Complexity of Hybrid Fusion: The patch-level fusion of visual and geometric features, along with the mixed positional embedding, adds complexity to the model architecture, potentially increasing the risk of overfitting or making it harder to fine-tune.
4. Limited Benchmarking: Although the method performs well on the RGBD-3DMatch, RGBD-3DLoMatch and Kitti datasets, it may not generalize as effectively to other types of point cloud registration tasks or datasets not covered in the experiments.

**Questions:**

Here are several questions based on the directions you provided:

1.Paradigm Shift: The introduction of image information into point cloud registration represents a significant shift in the paradigm. While this is an innovative approach, it also makes traditional point cloud registration methods, which were used for comparison, seem outdated in comparison. Do you think this shift is essential for the future of point cloud registration, or could there be other approaches that maintain the previous paradigm while improving performance?

2.Computational Complexity: While integrating DINOv2 to extract features from images is a promising idea, it undeniably increases the computational complexity. You briefly mention the time performance, but we have some concerns regarding the computational cost. Could you provide a more precise evaluation of the model's complexity, such as FLOPS, memory usage, or execution time in a more detailed manner?

3.Feature Fusion and Transformer Modifications: The feature fusion and transformer modifications you introduced are interesting, but they seem to follow relatively common practices in the field. We appreciate the creativity, but do you believe there are more novel paradigms or more robust methods that could be explored for this task? Could deeper, more innovative thinking lead to even better results or more groundbreaking contributions in this area? We're looking forward to it.

---

### Official Review · Reviewer_jibn · 2025-10-30

**Soundness:** 3
**Presentation:** 3
**Contribution:** 3
**Rating:** 4
**Confidence:** 5

**Summary:**

In this paper, the authors propose DINOReg, a point cloud registration network that fully leverages both visual and geometric information by extracting visual features with DINOv2 and fusing them with geometric features at the patch level. Extensive experiments on RGBD-3DMatch and RGBD-3DLoMatch demonstrate that DINOReg significantly outperforms state-of-the-art geometry-only and multi-modal methods, achieving a 14.2% increase in patch inlier ratio and a 15.7% increase in registration recall.

**Strengths:**

(1) The paper is well-organized and easy to follow.

(2) This work constructs the RGBD-3DMatch & RGBD-3DLoMatch datasets to evaluate point cloud registration methods using image–point cloud pair data.

(3) The proposed DINOReg achieves SOTA performance in point cloud registration.

**Weaknesses:**

(1) The rationale for using vision foundation models like DINOv2 is based on their general feature extraction capabilities; however, the paper does not provide evidence that these features are specifically advantageous for point cloud registration, making the motivation somewhat speculative.

(2) The proposed mixed positional embedding, while potentially useful, may be considered a relatively minor technical improvement and does not represent a major conceptual or methodological breakthrough in point cloud registration.

(3) Several closely related works are missing from the related work, such as [1] and [2].

(4) The spatial mapping relies heavily on accurate camera calibration. Any errors in intrinsic or extrinsic parameters could result in misalignment between visual and geometric patches, degrading the quality of feature fusion.

(5) The robustness evaluation only considers Gaussian noise with σ = 5 or 10, which may not fully reflect real-world mapping inaccuracies such as nonlinear distortions, occlusions, or large viewpoint variations.

(6) Visualized results are not provided, which weakens the reliability and interpretability of the quantitative results.

(7) The effect of the window size in the window aggregation module remains unclear. More insights into the choice of kernel size (3×3) would be helpful.

[1] Wang Z, Huang S, Butt J A, et al. Cross-modal feature fusion for robust point cloud registration with ambiguous geometry[J]. ISPRS Journal of Photogrammetry and Remote Sensing, 2025, 227: 31-47.

[2] X Kang, Z Luan, K Khoshelham,, et al. Equi-gspr: Equivariant se (3) graph network model for sparse point cloud registration. ECCV 2025

**Questions:**

(1) How does the proposed mixed positional embedding qualitatively improve registration compared to existing positional encoding methods?

(2) How does the method perform under more realistic conditions, such as nonlinear distortions, occlusions, sparse point clouds, or large viewpoint differences?

(3) Could the authors explore or provide evidence on whether other vision foundation models, aside from DINOv2, can effectively contribute to multi-modal point cloud registration, and how their performance compares?

(4) Can the authors provide more insights and qualitative visualizations to support the quantitative results and validate design decisions of kernel size in window aggregation module?

---

### Official Review · Reviewer_iGf1 · 2025-10-31

**Soundness:** 2
**Presentation:** 2
**Contribution:** 1
**Rating:** 2
**Confidence:** 5

**Summary:**

The manuscript introduces DINOReg, a  framework for point cloud registration that integrates both visual and geometric information. The authors claim that combining visual and geometric cues allows the model to achieve more robust and reliable registration performance.
In summary, I believe the main idea of the paper to include visual info as additional clues is not novel. Which has been widely used, especially in image-point cloud registration problems. Adapting this concept to pure point cloud registration does not represent a substantial methodological innovation.

**Strengths:**

1.The paper attempts to exploit visual cues in addition to geometric data for point cloud registration, which is a reasonable and potentially effective direction.

2. The adoption of DINOv2, a strong visual representation model, reflects an effort to leverage recent advances in self-supervised vision transformers,  which is also a reasonable and potentially effective direction.

**Weaknesses:**

1.The core idea lacks novelty. The inclusion of visual information as auxiliary cues has been extensively explored in prior literature, particularly in image–point cloud registration tasks. Adapting this concept to pure point cloud registration does not represent a substantial methodological innovation.

2.The paper does not provide any visual qualitative results, either in the main body or the appendix. This omission significantly limits the ability to evaluate the qualitative effectiveness and visual interpretability of the proposed method.

3. Several key related works are missing, most notably DIFF-Reg, which represents a state-of-the-art approach for 3D non-rigid registration. The absence of such comparisons weakens the contextual grounding and completeness of the literature review.

4. The paper lacks runtime and computational efficiency analysis, making it difficult to assess the practical feasibility and scalability of the proposed method in real-world scenarios.

**Questions:**

Please refer to weakness

---

### Official Review · Reviewer_PSDE · 2025-11-03

**Soundness:** 3
**Presentation:** 2
**Contribution:** 2
**Rating:** 4
**Confidence:** 4

**Summary:**

The paper proposes DINOReg, a point cloud registration network that effectively fuses visual information from images with geometric information from point clouds. The core idea is to leverage a powerful Vision Foundation Model (DINOv2) to extract dense texture and semantic features from images, which are then fused with local geometric features from a 3D backbone at the patch level. This multi-modal fusion is enhanced by a Mixed Positional Embedding that incorporates spatial relationships from both 2D image space and 3D point cloud space, leading to more distinctive and robust feature representations for matching challenging point cloud pairs.

**Strengths:**

The main contributions include:

The paper claims to be the first to utilize a Vision Foundation Model for Registration, which introduces DINOv2 as a visual backbone to extract powerful, general-purpose features that contain rich texture and global semantic context, thereby moving beyond simple color information.

The paper proposes Patch-Level Multi-Modal Fusion, which proposes a spatial mapping and window aggregation strategy to align and fuse visual (DINOv2) and geometric (KPConv-FPN) features at the patch level. This mitigates mapping inaccuracies and avoids the information loss common in point-wise fusion methods.

The paper designs a mixed positional embedding that injects relative positional information from both 2D pixel coordinates and 3D geometric structures into the self-attention mechanism. This enhances the model's ability to perceive spatial relationships.

The paper constructs the RGBD-3DMatch and RGBD-3DLoMatch datasets from the original 3DMatch scans, providing a dedicated benchmark for image-point cloud registration that is more challenging and realistic than previous datasets.

**Weaknesses:**

Utilizing a large foundation model (DINOv2) increases the computational cost and inference time compared to geometry-only methods such as GeoTransformer, which should be discussed.

Limited visual comparisons. Although the metrics are getting a little better, however, it is hard to evaluate the performance boost without visualizing the comparisons. I suggest adding more visual comparisons.

The method is designed for scenarios where paired image and point cloud data are available (e.g., from RGB-D cameras or calibrated camera-LiDAR systems). While fusing multi-modal inputs are good, it is not directly applicable to pure point cloud registration tasks without corresponding images.

The performance is tied to the capabilities of the pre-trained DINOv2 model. While this is generally beneficial, it may inherit any biases or limitations from its original training data. Which also limits the new contributions of the paper.

**Questions:**

Please respond to ``Weaknesses''.

---

### Note · Authors · 2025-11-15

**Comment:**

We sincerely thank the reviewers for their valuable comments. We appreciate your positive comments regarding the motivation and performance. And we also appreciate the weaknesses and concerns you pointed out.

We believe our research can bring new insights to multi-modal point cloud registration, as we leverage image information in a novel manner, which demonstrates significant advantages compared to existing methods. However, we acknowledge that there are additional experiments and analysis needed for further demonstration, such as qualitative results, complexity analysis and evaluation using other visual pretrained models. Therefore, we decide to withdraw the manuscript, and we will thoroughly revise the manuscript based on the feedback.

Thank you again for your time and constructive feedback!

**Withdrawal Confirmation:**

I have read and agree with the venue's withdrawal policy on behalf of myself and my co-authors.